# Carbapenem-Resistant NDM and OXA-48-*like* Producing *K. pneumoniae*: From Menacing Superbug to a Mundane Bacteria; A Retrospective Study in a Romanian Tertiary Hospital

**DOI:** 10.3390/antibiotics13050435

**Published:** 2024-05-12

**Authors:** Dragos Stefan Lazar, Maria Nica, Amalia Dascalu, Corina Oprisan, Oana Albu, Daniel Romeo Codreanu, Alma Gabriela Kosa, Corneliu Petru Popescu, Simin Aysel Florescu

**Affiliations:** 1“Dr. Victor Babes” Clinical Hospital of Infectious and Tropical Diseases, “Carol Davila” University of Medicine and Pharmacy, 050474 Bucharest, Romania; maria.nica@umfcd.ro (M.N.); corneliu.popescu@umfcd.ro (C.P.P.); 2“Dr. Victor Babes” Clinical Hospital of Infectious and Tropical Diseases, 030303 Bucharest, Romania; amalia.dascalu@spitalulbabes.ro (A.D.); corina.oprisan@spitalulbabes.ro (C.O.); claudia.albu@spitalulbabes.ro (O.A.); daniel.chirita@spitalulbabes.ro (D.R.C.); alma.kosa@spitalulbabes.ro (A.G.K.)

**Keywords:** Carbapenem-resistant *Klebsiella pneumoniae*, carbapenemases, ESKAPE pathogens, NDM+OXA-48-*like*, antibiotic therapies

## Abstract

Background: Carbapenem-resistant *Klebsiella pneumoniae* (Cr-Kpn) is becoming a growing public health problem through the failure of adequate treatment. This study’s objectives are to describe the sources of Cr-Kpn in our hospital over 22 months, associating factors with the outcome of Cr-Kpn-positive patients, especially those with NDM+OXA-48-*like* (New Delhi Metallo-β-Lactamase and oxacillinase-48)*,* and the effectiveness of the treatments used. Methods: A retrospective observational cohort study including all hospitalized patients with Cr-Kpn isolates. We reported data as percentages and identified independent predictors for mortality over hospital time through multivariate analysis. Results: The main type of carbapenemases identified were NDM+OXA-48-*like* (49.4%). The statistical analysis identified that diabetes and co-infections with the Gram-negative, non-urinary sites of infection were factors of unfavorable evolution. The Cox regression model identified factors associated with a poor outcome: ICU admission (HR of 2.38), previous medical wards transition (HR of 4.69), and carbapenemase type NDM (HR of 5.98). We did not find the superiority of an antibiotic regimen, especially in the case of NDM+OXA-48-*like*. Conclusions: The increase in the incidence of Cr-Kpn infections, especially with NDM+OXA-48-*like* pathogens, requires a paradigm shift in both the treatment of infected patients and the control of the spread of these pathogens, which calls for a change in public health policy regarding the use of antibiotics and the pursuit of a One Health approach.

## 1. Introduction

*K. pneumoniae* (Kpn), belonging to the Enterobacterales order, is a bacterium widely distributed in nature, commonly residing in the intestinal tracts of both animals and humans, and also present in various environmental niches like water, food, and soil [1,2]. It exhibits opportunistic behavior, leading to medical conditions such as urinary tract infections, skin infections, pneumonia, sepsis, and other diseases associated with considerable morbidity and mortality [3,4], primarily attributed to its virulence and antibiotic resistance.

The discovery of antibiotic resistance dates back to the early days of treatment attempts. It has since escalated alongside the emergence of new antibiotic classes, with reports emerging from various regions worldwide [4]. Carbapenems once hailed as “super-antibiotics” upon their discovery, have undergone a rapid decline in effectiveness in recent decades due to the emergence of carbapenemases, enzymes present in several Gram-negative bacteria such as *K. pneumoniae*, *Acinetobacter baumannii*, *Pseudomonas aeruginosa*, and *Enterobacter* spp. These, along with two Gram-positive bacteria, *Enterococcus faecium,* and *Staphylococcus aureus*, collectively constitute a class of “super-bacteria” categorized by an acronym (ESKAPE), a term with dual significance reflecting both the first letter of the bacterial name and their ability to evade the bactericidal effects of antibiotics [5,6].

*K. pneumoniae* has demonstrated a remarkable capacity to rapidly develop resistance to commonly used antibiotics, reflecting its adaptive response to environmental changes. Carbapenem-resistant *K. pneumoniae* (Cr-Kpn) is encountered worldwide and has been identified as the pathogen with the fastest-growing antibiotic resistance in Europe [7,8]. Since 2020, it has been considered the primary pathogen causing hospital-acquired bacterial pneumonia [9]. Several mechanisms can explain its resistance to carbapenems. Thus, Cr-Kpn is capable of producing extended-spectrum β-lactamases (ESBL) with the ability to hydrolyze cephalosporins in combination with decreased permeability of the cell wall through modification of porins, which increases antibiotic resistance, as well as the upregulation of efflux mechanisms that reduce intrabacterial antibiotic concentration. Additionally, it can develop resistance to aminoglycosides through ribosomal mutations and mutations in the quinolone resistance-determining region (QRDR) [10]. These mechanisms create a strong capacity for resistance to most classes of antibiotics.

Carbapenemases are a group of β-lactamases that hydrolyze carbapenems. The Bush–Jacoby–Medeiros classification system [11] categorizes β-lactamases according to their properties, in particular substrate and inhibitor profiles. This system comprises four main groups: Group 1 (penicillins), Group 2 (extended-spectrum β-lactamases or ESBLs), Group 3 (carbapenemases) and Group 4 (inhibitor-resistant β-lactamases). Carbapenemases are divided into molecular classes A-D based on their structure according to the Ambler classification system [12]. Class A carbapenemases include enzymes such as KPC and IMI; class B includes metal β-lactamases such as NDM, IMP and VIM; class C includes enzymes encoded by the *amp*C gene, which is carried on the chromosome of many Enterobacteriaceae and acts as cephalosporins; and class D includes enzymes such as OXA-23, OXA-24/40, OXA-48 and OXA-58.

The most common carbapenemases in Kpn cases are KPC (*K. pneumoniae* carbapenemase), VIM (Verona integron-encoded metallo-β-lactamase), NDM (New Delhi metallo-β-lactamase), and OXA-48 (oxacillinase-48). Some derivatives of OXA-48 (OXA-181, OXA-204, and OXA-232) with similar hydrolytic properties have also been identified in *K. pneumoniae* (OXA-48-*like*) [13]. According to the Bush–Jacoby–Medeiros classification, KPC, NDM, and VIM belong to group 3, while OXA-48-*like* enzymes belong to group 2 (ESBLs that can hydrolyze carbapenems) [14].

Currently, resistance genes are thought to be transferred by mobile genetic elements (plasmids and transposons) both within and between bacterial species [15,16]. Kpn is a widely distributed bacterium in the environment and therefore gene exchange, including antibiotic resistance, can be considered natural. Most likely, the sources are primarily found in hospitals, where epidemic outbreaks have been frequently reported [17,18]. Cr-Kpn has been detected in the municipal environment in hospital wastewater [19] and in rivers and lakes [20]. Veterinarians have reported the presence of Cr-Kpn in domestic animals [21], wildlife [22], or animals intended for human consumption, such as pigs [23] or poultry [24]. The presence of Cr-Kpn has even been detected in domestic cockroaches [25] (OXA-48 and NDM), suggesting the possibility of transmission to humans via this route. These findings suggest a possible transmission of resistant germs or resistance genes from the environment, especially considering that carbapenems are banned in the treatment of animals. Consequently, the presence of Cr-Kpn in the environment could indicate a zoonotic source for the acquisition of carbapenem-resistant *K. pneumoniae* infections in humans.

The appearance of genes encoding Kpn carbapenemases has been reported since 1996 in the United States (KPC), in 2003 OXA-48 in isolates from Turkey, and NDM was discovered in 2008 in a patient in Sweden originating from India [14]. The widespread use of carbapenems in antibiotic treatments has led to endemic Cr-Kpn infections in certain areas of the globe (Latin America, Gulf countries, China, and Southeastern Europe) [14]. In Europe, there is a discrepancy in resistance depending on the region. According to the Surveillance Atlas of Infectious Diseases from 2022 (ECDC) [26], in Scandinavian countries, the proportion of resistant isolates is less than 1%, while in Italy, it is 24.9%; in Bulgaria, 47.3%, and in Romania, 47.8%. The highest percentage of resistant isolates is in Greece (72%).

In Romania, the first official reports date back to 2012 (13.7%), and in the following years, the value increased to a maximum of 54.5% in 2021. In a study published in 2015 [27], OXA-48 was the main carbapenemase encountered, followed by NDM and VIM. No NDM + OXA-48 combinations were reported. At the “Dr. Victor Babes” Hospital of Infectious and Tropical Diseases (Bucharest, Romania), where the data for this study were collected, the identification of carbapenemase-resistant strains has experienced significant dynamics since 2010, with the percentage of Cr-Kpn reaching 40% in 2021, with a slight decrease in 2022 to 34.8%. A significant modification of the encountered carbapenemases has been observed regarding the identified carbapenemases. While in 2018, OXA-48-*like* was predominant (63.43%), in 2021 and 2022, the highest frequency encountered was NDM+OXA-48-*like* (28.78, 26%).

After the first reports of the presence of NDM+OXA-48-*like* strains in the “Dr. Victor Babes” hospital (19 cases, 2021), the number of patients infected with K. pneumoniae carrying this carbapenemase apparently increased in the following years at the expense of those with OXA 48-like, posing increasingly difficult therapeutic management problems for clinicians. As an infectious disease-only hospital admitting patients from many medical specialties, it can be considered a sentinel hospital for many strains of bacteria found in human pathology. For this reason, we set out to study cases of patients infected with Cr-Kpn to determine the differences that occur depending on the type of carbapenemase encountered. We chose this approach because we found little evidence in the literature of a differential analysis of these carbapenemases in relation to patient characteristics and outcomes. We hypothesized that the simultaneous presence of two carbapenemases (NDM and OXA-48-*like*) might exacerbate the development of patients infected with Cr-KPN. To this end, we followed demographic and epidemiological factors as well as the evolution of patients during hospitalization and the impact of “real-life” therapeutic regimens based not only on the bacterial resistance detected but above all on the specificities of the patients (comorbidities, drug interference, side effects).

## 2. Results

We identified carbapenemases in 172 hospitalized patients (single isolates). However, characterization of the carbapenemases could not be achieved in ten of these cases, so this study included only 162 cases. The baseline characteristics are summarized in Table 1 below, where demographics and other clinically relevant data were assessed.

The distribution of patients with carbapenemases over the years is relatively similar: 80 patients were included in this study in 2022 and 82 patients in 2023. The value is slightly higher in 2023, even though we finalized the records two months earlier than the end of the year. 

The NDM+OXA-48-*like* variant carbapenemase encountered was the most prevalent in both years (48.8% in 2022 and 50% in 2023), followed by Oxa48 (23.8% in 2022 and 18.3% in 2023). KPC had 17 (21.3%) isolates in 2022 and 15 (18.3%) in 2023. The lowest numbers were found for NDM, with only 5 (6.3%) in 2022 and 11 (13.4%) in 2023.

Regarding the resistance pattern encountered, we have recorded higher sensitivities to ceftazidime-avibactam (CAZ) 44.3%, colistin (COL) 41.4%, amikacin (AMK) 22.4%, gentamicin (GEN)16.9%, trimethoprim-sulfamethoxazole (STX) 14%, with the rest of the tested antibiotics showing sensitivity levels below 5%, except for Tobramycin (TOB) being slightly over (6.9%) (Appendix A). 

Regarding the resistance found in NDM+OXA-48-*like* isolates, 33% of them were sensitive to colistin, 9% to trimethoprim-sulfamethoxazole, and only 8% to ceftazidime-avibactam; with the rest of the antibiotics, sensitivities <5% or absolute resistance are found (Appendix A).

In this study, we could not evaluate resistance to cefiderocol until 2023, using the microdilution method, according to EUCAST [28], the sensitivity level for Cr-Kpn isolates being 60%.

### 2.1. Characteristics of the Studied Cohort

The average age in the entire cohort was 67.4 years (95% CI 65.06–69.83), and there were no statistically significant differences between the four groups of carbapenem-resistant *K. pneumoniae* isolates.

Regarding the contribution of NDM + OXA 48-like carbapenemases in this study brought by the gender variable, we have 32.5% (%95 CI 17.09–45.79, *p* < 0.0001) more in men than in women. The rest of the carbapenemases (Oxa48, KPC, and NDM) do not show significant differences between sexes. 

The majority of patients, 131 (80.9%), came from urban areas, implying that the percentage of isolates with carbapenemase is much higher among them than among those from rural areas, which account only for 31 (19.1%) patients.

### 2.2. Previous Hospitalizations vs. Non-Hospitalized within 12 Months

Out of 162 patients for whom the type of carbapenemases could be identified, 80.9% had hospitalizations in other wards in the last 12 months. However, we considered 19.1% of those not hospitalized in the last year. They were infected significantly with carbapenemase-producing *K. pneumoniae*, suggesting the important circulation of this type of bacteria in the environment. Moreover, the level of NDM+OXA-48-*like* was higher in patients with previous hospitalizations (80%) versus only 20% in non-hospitalized patients. Also, KPC (71.9%), NDM (81.3%), and Oxa48 (91.2%) were in higher proportion in samples coming from patients who had previous hospitalization (Appendix A). 

### 2.3. Previous Hospitalizations According to Medical Specialties

We evaluated the types of carbapenemases encountered in patients with previous hospitalizations based on the type of medical specialties. Since there were patients with multiple hospitalizations in both surgical and medical specialties in the last 12 months, we divided these entities into patients with a surgical history, those with a medical history, and those without a hospitalization history. In the first group, we included patients who had at least one hospitalization in surgery, regardless of whether they had subsequent hospitalizations. In the second group, we included patients strictly hospitalized for medical conditions. 

Of the 131 patients who went through other hospitals, the most common carbapenemase was NDM+OXA-48-*like* (64, 48.9%); of them, 46 (71.9%) went through the medical wards, while only 18 (28.1%) went through the surgical wards. Furthermore, more isolates were found in patients treated in the medical ward than in the surgical ward (Oxa48-67.7%, NDM-53.8% and KPC-73.9%) (Appendix A). 

### 2.4. Comorbidities

Only those with a frequency of at least five cases in each category were analyzed. For this reason, only patients with neoplasms and diabetes mellitus were included in the analysis.

In case of patients with neoplasms 47 (29%), we found that the NDM+OXA-48-*like* phenotype was present in the majority of cases (51.1%), followed by KPC at 21.3%, OXA48 at 17%, and NDM with 10.6% of cases.

In case of diabetic patients 53 (32.7), we found that the NDM+OXA-48-*like* phenotype was present in the majority of cases with a significant percentage (50.9%), followed by OXA48 (18.9%), and both KPC and NDM having equal percentages (15.1%). 

## 3. Analysis of Hospitalized Patients

### 3.1. Distribution According to Clinical Forms of Illness at Admission (Mild, Moderate, Severe)

Most patients were classified as having moderate forms, 111 (68.5%). Among them, the most frequent phenotype was NDM+OXA-48-*like* (45%), followed by Oxa48 (24.3%) and KPC (20.7%). For severe/critical forms, 34 (21.0%) patients, the most common phenotype was NDM+OXA-48-*like* (61.8%). Also, for severe/critical patients, KPC and Oxa48 were present in relatively similar percentages (17.6%) (Figure 1).

Regarding admission to the intensive care unit (ICU), patients in the study group had a significantly higher percentage of NDM+OXA-48-*like* isolates (60%) compared to other isolates (17.1%KPC, 5.7%NDM and 17.1%Oxa48).

### 3.2. The presence of Carbapenemases in Urine Samples

Because carbapenem-resistant *K. pneumoniae* isolates were obtained primarily from urine samples 95 (58.6%), we compared patients with urinary tract infections (UTI) with the rest of the cohort (non-UTI) for the types of carbapenemases. We found no significant difference for NDM + OXA48-like (56.3 vs. 43.8%), for KPC a difference of 25% (62.5 vs. 37.5%), for NDM no significant difference and for Oxa48 more in ITU patients (61.8%) than in non-ITU patients (38.2%).

### 3.3. The Simultaneous Presence of Gram-Negative Bacteria

The possibility of resistance gene transfer among Enterobacterales is mentioned in the literature [29,30]. We analyzed whether their presence (both in proven coinfections and carriage) coincided with the presence of a specific type of carbapenemase. We analyzed the distribution of patients infected with Cr-Kpn based on the presence of other Gram-negative bacteria isolated from 79 bacteriological samples during hospitalization. The most frequently encountered were *Pseudomonas* spp., *Acinetobacter* spp., *E. coli*, *Proteus* spp., and *Providencia* spp. Combinations of Gram-negative bacteria were found in 28 patients. Gram-positive bacteria and fungi isolated during microbiological sampling were excluded from the statistical analysis.

In the case of OXA48, the most frequent co-occurrences were with *E. coli* and *Providencia* spp. in 50% of isolates. For NDM+OXA-48-*like*, co-occurrence was found with 66.8% of *Pseudomonas* spp. strains and 66.7% of *Proteus* spp. strains. Bacterial combinations had relatively equivalent distributions within the studied resistance classes (Figure 2).

## 4. Treatments

To assess the relevance of the administered antibiotic treatment, we excluded from the initial batch of 172 patients those who died within the first 72 h of hospitalization, considering that until then, bacteriological results were largely unavailable (7 patients), as well as four patients for whom the antibiotic treatment could not be well documented. Patients for whom the type of carbapenemase could not be characterized were also excluded (7 patients).

Meropenem, ceftazidime-avibactam, colistin, levofloxacin, tigecycline, and trimethoprim-sulfamethoxazole were used in monotherapy regimens. Antibiotic combinations prescribed included cephalosporins (ceftazidime, ceftriaxone, ceftazidime-avibactam), colistin, aminoglycosides (amikacin), fluoroquinolones (levofloxacin), tigecycline, chloramphenicol, trimethoprim-sulfamethoxazole, at doses recommended by the hospital’s internal protocols and international guidelines. The doses were adjusted based on creatinine clearance [31]. The decision to administer antibiotics was made in most cases by attending physicians, all of whom were infectious disease specialists. The decision was made by consulting the internal stewardship committee in therapeutic impasse situations. Since there were numerous cases where therapeutic regimens were modified due to the evolution of patients or biological parameters, only the final therapeutic regimens received during hospitalization were considered in this work. 

### 4.1. The Carbapenem Regimen (Appendix A)

Out of the remaining 153 patients, we excluded the combination of meropenem + ceftazidime-avibactam bi-therapy (4 cases). A total of 72 patients received carbapenem-based regimens (meropenem at a dose of 2 g every 8 h), both in monotherapy and in combination therapy (with 3 or 4 antibiotics). A total of 77 patients received Carbapenem-sparing regimens in monotherapy or combination therapy. 

The treatment progress of patients with a carbapenem-based regimen was further analyzed, starting from the hypothesis that this combination of carbapenemases could negatively influence disease progression. In the table, we included subsets of patients with NDM+OXA-48-*like* carbapenemases (recovered/deceased) to compare with the rest.

We can conclude that in the treatment with meropenem, two out of three deaths occurred in patients with NDM+OXA-48-*like*. From the data analysis, it was noted that the third death occurred in an NDM+ patient, strengthening the recommendation not to use carbapenems as monotherapy in patients infected with NDM-positive *K. pneumonia* [30]. From a statistical point of view, no differences were observed between patients in the NDM+OXA-48-*like* subset and the rest of the carbapenemase-producing *K. pneumoniae* (Cr-Kpn) patients. 

Also, in this case, we did not observe statistically significant differences between the types of antibiotic therapies used and the final status of the patients (recovered/deceased). In 62 cases, we had evidence of infection resolution following treatment (clinical, biological, and bacteriological), out of which 33 cases were NDM+OXA-48-*like*. We excluded patients who showed evidence of *K. pneumoniae* carriage at discharge and those for whom bacteriological clearance could not be proven. 

In this situation, we analyzed treatments (both monotherapies and combination therapies) in order of their frequency of use to evaluate the achievement of significant recovery rates. We did not find statistically significant differences in favor of either monotherapy or combination antibiotic therapy among recovered or deceased patients. We found no statistically significant differences in favor of monotherapy or combination therapy with antibiotics in recovered or deceased patients. Likewise, no statistically significant differences were observed in patients within the NDM+OXA-48-*like* subgroup compared to those treated with carbapenem-based and carbapenem-sparing regimens.

### 4.2. The Ceftazidim-Avibactam Regimen (Appendix A)

The number of patients who received antibiotics with ceftazidime-avibactam (CZA) (in monotherapy or combined therapy) was 43. The number of patients who received CZA-sparing therapies was 115. We also excluded the patients who received meronem + CZA combined therapy from this group. We did not notice significant differences between the number of those cured who received monotherapy or combined therapies, which we also observed in the case of deaths. Even in the case of the NDM+OXA-48-*like* phenotype, we did not identify important differences regarding the cure or death rates. Looking more broadly at this therapy regarding survivors/deaths through bivariate analysis, we did not find statistically significant differences between CZA-based regimes and CZA spared regimes.

### 4.3. The Colistin Regimen (Appendix A)

In the case of those who received colistin in monotherapy or various combined regimens (n = 76), compared to colistin spared therapies (n = 77). We found a statistical difference between the cure rates favoring combined colistin-based therapies (95% CI 16.84% to 58.66% *p* = 0.0011). The burden of deaths was also more important in the case of patients treated with colistin based (95% CI 43.67% to 83.25%, *p* < 0.0001); regarding the NDM+OXA-48-*like* subgroup (n = 45), we found the same profile both in the case of cured patients (95% CI 3.09% to 56.83%, *p* = 0.0276) and those who died (95% CI 29.94% to 86.09%, *p* = 0.0007). In terms of survivors/deaths by bivariate analysis we found a statistically significant difference in favor of colistin-spared therapies (*p* = 0.008).

### 4.4. Patient’s Outcome

We assessed patients’ progression through bivariate analysis concerning carbapenemases and other demographic and clinical factors. We calculated the relative risk [RR] for variables found to be statistically significant in association with the outcome. All patients with less than three days of hospital stay were excluded from the analysis. 

From the bivariate analysis (results centralized in Appendix A), 8 out of 18 variables investigated (disease severity, admission to ICU, type of sample, bacterial coinfection, DM (diabetes mellitus), passage through urology wards and UTI (urinary tract infection) were statistically significantly associated with the clinical evolution of patients. Among these significant factors, particular interest lies in the severe/critical condition at admission of the patient, which has a [RR] of 7.50(95% CI 1.08–51.84) compared to mild condition. Although statistically insignificant, the moderate condition also exhibits a high [RR] value of 4.74 (95% CI 0.66–32.33). Admission to ICU is noteworthy due to its consistent [RR] of 2.92 (95% CI 1.91–4.47), bacterial coinfection presented a [RR] of 1.75 (95% CI 1.09–2.81) and diabetes mellitus condition also had a strong association with non-surviving [RR] 1.89 (95% CI 1.19–2.99).

The remaining variables associated with statistical significance displayed increased RR in the reference categories rather than the categories of interest. Thus, the type of sample from sources other than urine shows an [RR] of 2.2 (95% CI 1.37–3.55, *p* = 0.0011). Similarly, in the UTI variable, relative risk [RR] is higher in those without UTI [RR] 2.12 (95% CI 1.33–3.3, *p* = 0.0014).

### 4.5. Carbapenemase Influence on the Probability of Patient’s Survival

Survival distribution over the time being hospitalized (Kaplan–Meier survival analysis) indicates that the patients with NDM carbapenemases present had the lowest estimated median time of probability of survival, 28 days (CI95% was not calculated) compared to the rest of carbapenemases, 56 days (CI95% 46.42–65.57) for KPC and 46 days (CI95% 37.52–54.47) for NDM+OXA-48-*like*. No median survival time was calculated for Oxa48+ because its cumulative probability of survival does not go under 0.5 (Figure 3). Further, pairwise comparison was performed, and the long rank test indicates differences with statistical significance between NDM and KPC (*p* = 0.010) and NDM and NDM+OXA-48-*like* (*p* = 0.019).

### 4.6. Other Factors Influencing Patient’s Probability of Survival

We conducted a multivariate Cox regression analysis to assess what factors independently have a statistically significant impact on patient survival. This analysis included demographic variables (age and gender) as well as clinical variables that may impact patient outcomes such as admission to intensive care, type of sample, type of carbapenemases, presence of MDR coinfection, urinary tract infection, transition from other hospital wards, and comorbidities, and treatment strategy. 

From the Cox regression model, we identified factors associated with a probability of a poor outcome: age, with a hazard ratio [HR] of 1.06 (95% confidence interval CI 1.02–1.10); admission in ICU [HR] of 2.38 (95% confidence interval CI 1.07–5.29); carbapenemase type NDM, with an [HR] of 5.98 (95% CI 1.59–22.36); carriage of Gram-negative bacteria, with an HR of 1.91 (95% CI 1.03–3.98); previous transit in other medical specialties than surgery, with [HR] of 4.69 (95% CI 1.45–15.13). Notably, among carbapenemases types, NDM remains a significant factor associated with the likelihood of non-survival in patients. The urinary tract infection seems to be a „protective” factor (negative value of coefficient) with an [HR] of 0.172 (95% CI 0.060–0.493). Significant factors and their HR are centralized in the Table 2.

## 5. Discussion

The present study evaluated 172 patients admitted between January 2022 and October 2023, from which stems of Cr-Kpn were isolated. In the patients evaluated, we found an increase in the number of Cr-Kpn isolates in the year 2023 compared to 2022. In addition, NDM+OXA-48-*like* isolates experienced the same dynamics, becoming the most commonly found, 49.4% of the total batch studied. In the Infectious and Tropical Diseases Hospital “Dr. Victor Babes”, Bucharest, the first isolates in this category were in 2021 (19 cases). In 2022, 39 cases were isolated, and in the first ten months of 2023, 41 cases were recorded, which emphasizes a positive dynamic.

A higher sensitivity to ceftazidime-avibactam of 44.3% and colistin of 41.4% was observed in all strains (172). In Kpn NDM+OXA-48-*like* strains, sensitivity to antibiotics was lower, with the highest sensitivity to colistin at 32%, and ceftazidime-avibactam showed a sensitivity of 8%. For cefiderocol, data were available only for 2023, and sensitivity was 60%. This resistance level is worrying, especially concerning cefiderocol, which has not yet been introduced in Romania and showed an important level of resistance. 

The distribution of patients by age showed no significant changes. In the case of men, as well as in patients from the urban environment, an increased number of NDM+OXA-48-*like* was found, the differences being statistically significant.

Analyzing the conditions before the admission of these patients, 80.9% had admission up to 12 months before the evaluation in this study; most of them (48.90%) were with the NDM+OXA-48-*like* phenotype. We consider it important that among patients without previous hospitalizations (19.1%), a majority of 51.6% had NDM+OXA-48-*like* symptoms. This suggests a possible significant spread of this type of carbapenemases in the non-clinical setting, with important implications for the future. 

This resistance profile was found predominantly in patients who had transited medical specialties and had previous hospitalizations. The main carbapenemase encountered in both cases was NDM+OXA-48-*like*, without significant statistical differences.

Regarding comorbidities, in the most important analysis (neoplasms and diabetes mellitus), we also found a majority of NDM+OXA-48-*like* carbapenemases (more than 50%). The predominance of cases with NDM+OXA-48-*like* added to the immune status altered by the presence of diseases reveals the possibility of major difficulties in treating infection with *K. pneumoniae* in these two important subpopulations of patients.

Regarding the clinical forms that patients presented at hospitalization, the most common phenotype encountered in those with severe forms was NDM+OXA-48-*like* (61.8%), suggesting that it could generate more serious forms of the disease. A similar percentage (60%) was found in patients who needed intensive care (ICU), where the exacerbations may be due, in some cases, to the therapeutic inefficiency of the antibiotics used.

The presence in the same patient of other Gram-negative bacilli (coinfections or carriers), found in 79 cases, highlighted concomitant in the case of NDM+OXA-48-*like* with 66.8% of the strains of *Pseudomonas* spp. and 66.7% of *Proteus* spp., without finding statistically significant evidence. In the case of OXA-48, concomitant *E. coli* and *Providencia* spp. were found in 50% of cases (*p* = 0.0476).

With regard to antibacterial therapies, although monotherapy with carbapenems or therapeutic strategies that also contain these antibiotics are considered ineffective by many authors [30,31], in the case of patients infected with Kpn with NDM+OXA-48-*like* we have not encountered statistically significant differences between the carbapenemase-based regimen and the carbapenemase-sparing regimen. 

Even about the treatment regimens based on ceftazidime-avibactam, in comparison with other treatment regimens, we did not find significant differences in terms of bacteriological cure or the evolution towards the death of these patients. 

When examining the difference between colistin-based therapies and those where colistin was not used, we found that there were significant statistical differences in cure rates in favor of colistin-based therapies, but also in mortality rates when combined therapies were used. This possible paradox can be explained by the addition of toxicity brought by other antibiotics to colistin, an antibiotic with an important intrinsic toxicity. In addition, this type of therapy was used in the case of patients with more severe forms of the disease, with a higher probability of evolution towards death.

Nor was it possible to identify statistically quantifiable advantages or disadvantages concerning any therapeutic scheme used. In terms of treatment regimens, our study confirms some findings from the literature, which did not state that some treatment regimens can be recommended for Cr-Kpn infections in the presence of NDM or NDM+OXA-48-*like* phenotype [32,33,34]. An obvious limitation of this conclusion is given by the small number of patients who received the same therapeutic schemes and the unavailability of treatment with cefiderocol and aztreonam in the groups of patients studied.

Based on our bivariate analysis of the patients we studied, we found that diabetes, the condition upon admission, the presence of Gram-negative bacteria, the existence of Cr-Kpn in isolates other than those from the urinary tract, and ICU admission can negatively impact a patient’s prognosis. Through multivariate Cox regression analysis, we identified several factors independently associated with an increased likelihood of a poor outcome. These factors include age, the presence of New Delhi Metallo-beta-lactamase (NDM), carriage of Gram-negative bacteria, and previous medical specialty experience other than surgery. In a similar study comparing patients with Cr-Kpn infections with non-Cr-Kpn patients, without differentiating between carbapenemase types, it was found that independent factors for poor prognosis were: intensive care unit (ICU) stay, length of stay in hospital, antibiotic days of therapy >15 and previous carbapenem exposure [35].

Using Kaplan–Meier survival analysis, we determined that patients with NDM carbapenemases present had the lowest estimated median survival probability (28 days). Furthermore, our cohort analysis showed that the presence of the NDM+OXA-48-*like* phenotype was not an additional unfavorable factor for patient prognosis compared to other carbapenemases. Still, it was found in most of the subgroups of patients that we made up in this study.

### Strength and Limitation

Strengths: We consider this to be one of the first reports in Romania on the types of carbapenemases in patients hospitalized with Cr-Kpn, with a significant percentage of NDM+OXA-48-*like* strains detected. The results of this study did not show a more severe course in patients infected with NDM+OXA-48-*like* strains compared to patients infected with strains producing carbapenemases of other types. This study raises concerns about treatment regimens used in countries where access to antibiotic therapies such as cefiderocol or aztreonam is not yet available, as these regimens appear to have limited efficacy. 

Limitation of This study: This study was conducted in a hospital exclusively treating patients with infectious diseases, and the results obtained cannot be extrapolated to general hospitals. A significant number of patients were transferred from other medical specialties, and, in many cases, their bacteriological traceability (carrier or infected) could not be identified. The relatively small sample size of patients infected with Cr-Kpn resulted in subgroups with a small number of cases with NDM and KPC carbapenemases, which limited the statistical power of the analysis. The diverse treatment regimens resulted in heterogeneous patient groups, which are sometimes difficult to interpret statistically. For this reason, we believe that these results cannot be generalized and only apply to the sample studied.

## 6. Materials and Methods

We conducted a retrospective observational cohort study over 22 months, from January 2022 to October 2023. This study took place in ”Dr Victor Babes” Clinical Hospital of Infectious and Tropical Diseases, Bucharest, Romania. Biological samples were processed in the hospital’s microbiology laboratory.

The inclusion criteria are: adult patients (over 18 years old) hospitalized with infection with carbapenemase-producing *K. pneumoniae* as unique isolates. The exclusion criteria are: patients with proven Cr-Kpn carriage or infection with unclassified carbapenemases. In the subanalysis related to antibiotic treatments, we additionally excluded patients who died within the first 72 h of admission, as we considered that the etiology was not established until then.

Since the presence of carbapenemases is not an intrinsic virulence factor, the present study did not use an external control group, which could have been patients infected with other types of *K. pneumoniae*. Instead, we wanted to show the differences between patients infected with NDM+OXA-48-*like* strains and the other types of carbapenemases (KPC, NDM, OXA-48-*like*).

For patients with Klebsiella-positive isolates, from their in-hospital medical records, we extracted demographic aspects (age, sex, place of origin), medical history (comorbidities, transit through other hospitals and respective departments), and clinical aspects (clinical severity, type of infection, coinfections, MDR carrier status, treatment, and outcome). The severity of the cases was determined based on clinical, paraclinical, and laboratory data, with cases admitted to the intensive care unit (ICU) being classified as severe.

The collection and processing of biological samples was carried out by qualified, trained medical staff, and consisted of blood, urine, sputum, and bronchial aspirate samples, taking into account the presence of a single strain. The procedures were performed in consonance with the internal Standard Operating Procedures (SOP’s).

### 6.1. Identification of Carbapenemase-Producing K. pneumoniae Strains

The culture of biological samples was performed by classical methods, using dehydrated culture media or ready-made plates (Blood Agar Base, C.L.E.D. Medium, MacConkey Agar/OXOID, Basingstoke, UK) and for blood culture the BacT/ALERT automatic system. Identification of bacterial species was performed with automated identification systems based on the chemical principle (VITEK2C, VITEK2GN Biomerieux, Marcy-l’Étoile, France) and the physical principle, mass spectrometry (MALDI- TOF/Matrix- Assisted Laser Desorption/Ionization—Time of Flight Mass Spectrometry/BRUKER, Billerica, MA, USA).

Screening of bacterial strains for membership in a particular antibiotic resistance phenotype was performed using chromogenic media, with results then confirmed by standard methods.

Antibiotic sensitivity testing (antibiogram) was conducted using phenotypic techniques. Routine antibiotic sensitivity testing included disc diffusion (standardized disk diffusion method/Kirby Bauer), quantitative methods (MIC) in automated VITEK2C systems, concentration gradient method (E-Test), microdilution method in Mueller–Hinton broth (International Standard Reference (ISO) for non-fastidious aerobic bacteria/EUCAST). Bacterial isolate testing and interpretation of test results were performed following EUCAST Standard recommendations [36].

At the culture stage, for the isolation of ESBL or carbapenemase-producing *K. pneumoniae* strains, and to shorten the release time, we used chromogenic culture media (Brilliance ESBL Agar/Thermo Scientific (Waltham, MA, USA), Brilliance CRE Medium/OXOID).

Identification of carbapenemase types was performed with rapid tests (immunochromatographic principle/O.K.N.V.I. Resist-5/Coris BioConcept, Gembloux, Belgium), allowing identification of OXA48, KPC, NDM, VIM carbapenemases. Phenotypic results were confirmed by RT-PCR (GeneXpert/Xpert Carba-R/Cepheid, Singapore), a qualitative in vitro diagnostic method for rapid and differential detection of gene sequences associated with resistance of Gram-negative bacteria to carbapenems, such as *blaKPC*, *blaNDM*, *blaVIM*, *blaOXA-48* and *blaIMP-1* [37].

### 6.2. Statistical Analysis

Patient data related to demographics, medical history, clinical and paraclinical aspects, treatment length of stay in the hospital, and outcome were collected from the hospital informatics system and stored in an Office Excel (version 2021) dataset for further processing. 

Univariate analysis was performed using the Shapiro–Wilk test and the Kolmogorov–Smirnov test to assess the normality of continuous data. In bivariate analysis, to determine significant differences between categories, ANOVA tests were employed for normally distributed data, while the Mann–Whitney U test was used for non-normally distributed data. To assess the association between two categorical variables, Pearson’s chi-squared test and Yates’ correction for a 2 × 2 table or Fisher’s exact test for a table with more than 2 × 2 categories were utilized.

Additionally, the N-1 chi-squared test and the z-test for two-sample proportions were applied to check for significant differences in proportions. Also for the relative risk we used a chi-squared test and z-score for 95% CI when necessary (small sample data).

Kaplan–Meier survival analysis was conducted to assess the probability of survival beyond specific time points regarding the presence of carbapenemases. Survival curves for different carbapenemases were compared using the log-rank test, Breslow test, and Tarone Ware test.

To identify the independent factors, we conducted a multivariate Cox proportional hazards model when adjusted for other factors on patient outcomes (Cox regression). A significance cut-off level of 5% was utilized for all statistical tests performed. No imputation for missing data was applied and adjustment for confounding factors was performed in multiple regression analysis, controlling for all variables simultaneously. Statistical software IBM SPSS Statistics v26 and MedCalc (https://www.medcalc.org/calc/, accessed on 25 February 2024) were used for data processing.

## 7. Conclusions

*K. pneumoniae* infections resistant to carbapenemase have passed in recent years from the threat stage [38] to the usual presence, and the acquisition of additional resistance genes, as in the case of the NDM+OXA-48-*like* phenotype, only aggravates this situation. The presence of these “superbugs” not only in the hospital environment and the obvious difficulties in the therapeutic management of infected patients require a paradigm shift in the approach to Gram-negative germ infections. 

This study compared the characteristics and evolution of the patients, having as the main discriminator the type of carbapenemase encountered due to the appearance of NDM+OXA-48-*like* shifted in a short time from the stage of signaling in the literature to be regularly present. Even if these bacteria tend to become “mundane”, the infection implications are due to unreliable results despite numerous therapeutic approaches. In addition, the already reported resistance to new antibiotics (e.g., cefiderocol) is likely to jeopardize solutions in the near future.

The differences in antibiotic resistance of Cr-Kpn require new approaches. First, it is important to increase the number of microbiology laboratories able to determine different types of carbapenemases to avoid inappropriate treatments. Where this is not possible, screening for carbapenemases must be carried out, with bacterial strains subsequently sent to qualified laboratories. To reduce antimicrobial resistance, the implementation of antibiotic stewardship must be widespread in all these cases. To decrease antimicrobial resistance, it is necessary to spread the implementation of antibiotic stewardship in all these cases. 

Concerning the circulation of these bacteria in the environment, a One Health approach is required to assess their presence and limit their spread into the environment. This requires close collaboration between veterinary and human health professionals and environmental specialists. These approaches are necessary to quickly and efficiently identify the likelihood of resistance mechanisms occurring in a geographical area and to take coordinated and appropriate action.

We consider it essential that with the emergence of new antibiotics (aztreonam, cefiderocol, aztreonam/avibactam, etc.), further research is carried out to improve the evolution of patients and avoid unnecessary treatments.

## Figures and Tables

**Figure 1 antibiotics-13-00435-f001:**
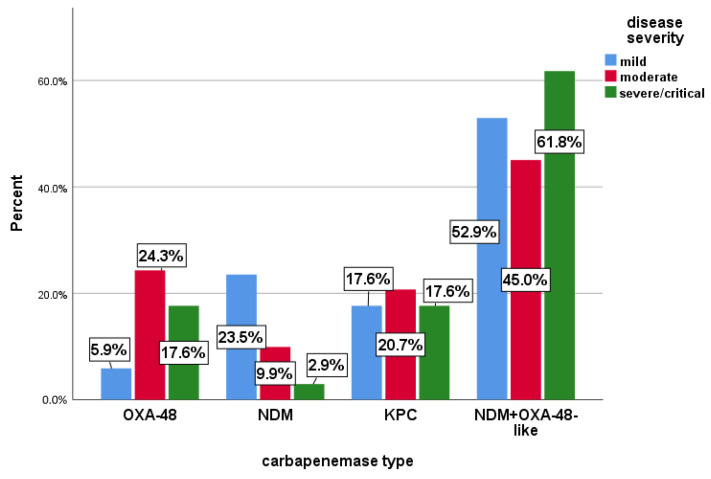
Distribution of Carbapenemases upon Hospital Admission Based on Disease Severity.

**Figure 2 antibiotics-13-00435-f002:**
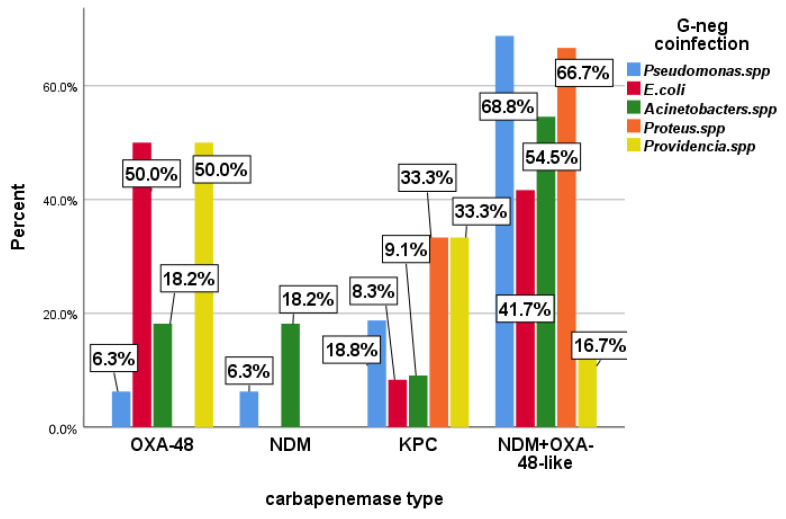
Association of Gram-Negative (G-neg) Germs Coinfection with Carbapenemases from Isolates.

**Figure 3 antibiotics-13-00435-f003:**
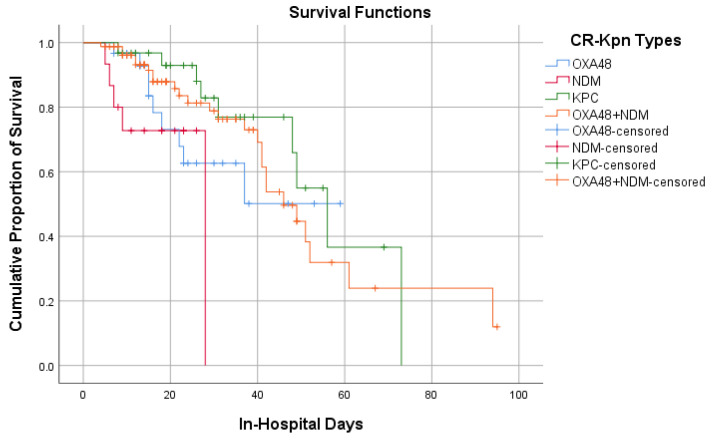
Probability of Survival Regarding Carbapenemase Types.

**Table 1 antibiotics-13-00435-t001:** Baseline Characteristics and Bivariate Analysis to Assess the Association of Variables with the Carbapenemases Isolated.

Variable	All n (%) n = 162	OXA 48 n = 34	NDM n = 16	KPC n = 32	NDM+OXA-48-*like* n = 80	*p* ^1^-Value
**Study year**						-
**2022**	80 (49.4)	19 (23.8)	5 (6.3)	17 (21.3)	39 (48.8)	-
**2023**	82 (50.6)	15 (18.3)	11 (13.4)	15 (18.3)	41 (50.0)	-
**Age (avg, SD)** ^2^	67.4 (15.37)	67.2 (15.5)	62.8 (15.6)	72.7 (11.9)	66.3 (16.1)	0.128
**Gender**						0.222
F	68 (42.0)	17 (25.0)	8 (11.8)	16 (23.5)	27 (39.7)	
M	94 (58.0)	17 (18.1)	8 (8.5)	16 (17.0)	53 (56.4)	
**Urban or rural origin**						0.457
Rural	31 (19.1)	7 (22.6)	5 (16.1)	7 (22.6)	12 (38.7)	
Urban	131 (80.9)	27 (20.6)	11 (8.4)	25 (19.1)	68 (51.9)	
**Type of infection**						0.906
non-UTI ^3^	67 (41.4)	13 (19.4)	7 (10.4)	12 (17.9)	35 (52.2)	
UTI	95 (58.6)	21 (22.0)	9 (9.5)	20 (21.1)	45 (47.4)	
**In-hospital days** **(avg, SD)**	25.8 (18.77)	20.7 (14.15)	15.06 (8.48)	30.6 (17.95)	27.0 (18.67)	0.04
**Disease severity at admission**						0.164
mild	17 (10.5)	1 (5.9)	4 (23.5)	3 (17.6)	9 (52.9)	
moderate	111 (68.5)	27 (24.3)	11 (9.9)	23 (20.7)	50 (45.0)	
severe/critical	34 (21.0)	6 (17.6)	1 (2.9)	6 (17.6)	21 (61.8)	
**ICU** ^4^ **admission, n (%)**	35 (21.6)	6 (17.1)	2 (5.7)	6 (17.1)	21 (60.0)	0.518
**Comorbidities n (%)**						
Neoplazia	47 (29.0)	8 (17.0)	5 (10.6)	10 (21.3)	24 (51.1)	0.885
Diabetes mellitus	53 (32.7)	10 (18.9)	8 (15.1)	8 (15.1)	27 (50.9)	0.355
**Bacterial coinfection, n (%)**	67 (41.3)	15 (22.4)	3 (4.5)	15 (22.4)	34 (50.7)	0.270
**MDR carriage, n (%)**	58 (35.8)	12 (20.7)	5 (8.6)	10 (17.2)	31 (53.4)	0.864
**G-negative bacteria coinfection and/or carriage**						0.097
*Pseudomonas* spp.	16 (31.4)	1 (6.3)	1 (6.3)	3 (18.8)	11 (68.8)	
*E. coli*	12 (23.5)	6 (50.0)	0 (0.0)	1 (8.3)	5 (41.7)	
*Acinetobacter* spp.	11 (21.6)	2 (18.2)	2 (18.2)	1 (9.1)	6 (54.5)	
*Proteus* spp.	6 (11.8)	0 (0.0)	0 (0.0)	2 (33.3)	4 (66.7)	
*Providencia* spp.	6 (11.8)	3 (50.0)	0 (0.0)	2 (33.3)	1 (16.7)	
**Previous hospitalization (within 12 months)**						0.427
Non-hospitalized	31 (19.1)	3 (9.7)	3 (9.7)	9 (29.0)	16 (51.6)	
Surgery wards	40 (24.7)	10 (25.0)	6 (15.0)	6 (15.0)	18 (45.0)	
Medical wards	91 (56.2)	21 (23.1)	7 (7.7)	17 (18.7)	46 (50.5)	
**Carbapenem-based regimen (n = 149)**						0.081
Carbapenem spared	77 (51.7)	22 (28.6)	8 (10.4)	16 (20.8)	31 (40.3)	
Carbapenem based	72 (48.3)	10 (13.9)	5 (6.9)	15 (20.8)	42 (58.3)	
**Ceft-Avi** ^5^**-based regimen (n = 149)**						0.058
Ceft-Avi spared	115 (77.2)	20 (17.4)	11 (9.6)	22 (19.1)	62 (53.9)	
Ceft-Avi based	34 (22.8)	12 (35.3)	2 (5.9)	9 (26.5)	11 (32.4)	
**Colistin-based regimen (n = 153)**						0.066
Colistin spared	77 (50.3)	21 (27)	6 (7)	18 (23.3)	32 (41.5)	
Colistin based	76 (49.7)	10 (13.0)	8 (10.5)	13 (17.1)	45 (59.2)	
**Outcome**						0.985
survived	112 (69.1)	23 (20.5)	11 (9.8)	23 (20.5)	55 (49.1)	
not-survived	50 (30.9)	11 (22.0)	5 (10.0)	9 (18.0)	25 (50.0)	

^1^ *p*-value relevant to consider for statistical significance is below 5%; ^2^ avg = average and SD = standard deviation; ^3^ UTI = urinary tract infection; ^4^ ICU = intensive care unit; ^5^ Ceft-Avi = ceftazidime-avibactam.

**Table 2 antibiotics-13-00435-t002:** Significant Cox Regression Factors.

Variable	HR	CI95% (Lower-Upper)	*p*-Value
Age	1.06	1.02–1.10	<0.0001
ICU admission	2.38	1.07–5.29	0.032
Transit in non-surgical specialties	4.69	1.45–15.13	0.010
Carbapenemase type: NDM	5.98	1.59–22.36	0.008

## Data Availability

The corresponding author can provide the data used in this study upon reasoned request.

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
