# Peer review of "Carbapenem-Resistant NDM and OXA-48-like Producing K. pneumoniae: From Menacing Superbug to a Mundane Bacteria; A Retrospective Study in a Romanian Tertiary Hospital"

_antibiotics, 2024, doi:10.3390/antibiotics13050435_

Round 1

Reviewer 1 Report

Comments and Suggestions for Authors

The authors submitted a manuscript titled "Carbapenem-resistant NDM and OXA 48-like producing K. pneumoniae: from Menacing Superbug to a Mundane Bacteria" for consideration in the journa. This study presents a retrospective observational cohort analysis over a period of 22 months, focusing on hospitalized patients with isolates of carbapenem-resistant Klebsiella pneumoniae (Cr-Kpn), specifically those harboring NDM and OXA-48-like carbapenemases. Please see my comments below. 

A statistically significant difference was observed with colistin-based therapies, which showed higher cure rates but also a higher incidence of death, attributed to the toxicity of combined therapies and the severe condition of patients treated with colistin. The study did not find a clear advantage for any therapeutic scheme, echoing the broader literature's uncertainty about the optimal treatment for Cr-Kpn infections, especially those involving NDM or NDM+OXA48-like enzymes. Limitations noted include the small sample sizes for specific treatment comparisons and the unavailability of newer treatments like cefiderocol and aztreonam. Additionally, factors such as diabetes, admission condition, presence of gram-negative bacteria, non-urinary Cr-Kpn isolates, and ICU admission were found to negatively impact prognosis.

Initial comments pertaining to the introduction section

-While the introductionn effectively sets up the problem statement regarding Cr-Kpn, it, could more clearly state the specific gap in the literature or in current understanding that this study aims to address.

-Recommend that study objectives are explicitly stated at the end of the introduction section. The direction of this manuscript is not clear after reading the introduction.

-Recommend a more accurate description of the ambler and/or functional classification of Bush-Jacobi when introducing carbapenemases. Zinc dependency is not overall pertinent to the classification from an introductory standpoint.

Control group:

- The lack of an external control group in a retrospective study can significantly limit the ability to draw causal inferences from the observed associations. Without a control group, it's challenging to determine whether the outcomes observed in patients with carbapenemase-producing Klebsiella pneumoniae infections are directly attributable to the bacteria's characteristics or resistance mechanisms, or if they're influenced by other confounding factors. After readingg this section, I am concerned that there is no mention of a control group or justification for the absence of a control group.

-For a study on antibiotic resistance, a control group could consist of patients infected with non-resistant strains of Klebsiella pneumoniae or another comparator that helps isolate the impact of carbapenemase production on clinical outcomes. The absence of such a group hampers the ability to clearly delineate the effects of resistance on treatment efficacy, morbidity, and mortality.

-If a control group is inherently lacking due to the study's design, the authors could consider using statistical methods to create matched cohorts within the available data. Techniques like propensity score matching can help to approximate the effect of having a control group by matching patients based on relevant covariates.

-With regards to the result section, the manuscript effectively presents a wide array of data, including demographics, antibiotic resistance patterns, and outcomes associated with different carbapenemase types.

-Absence of Control Group: The retrospective design and the lack of a control group limit the ability to draw causal inferences from the observed associations. For example, without comparing outcomes to patients infected with non-resistant strains or receiving different treatments, it's challenging to ascertain the specific impact of NDM and OXA-48-like enzymes on patient outcomes. This limitation should be explicitly acknowledged and discussed in relation to how it might affect the study's conclusions.

-Given the study's setting and its retrospective nature, the generalizability of the findings to other contexts or populations might be limited. This concern is particularly relevant given the variability in carbapenem-resistant K. pneumoniae prevalence and resistance mechanisms globally. The discussion should critically address the generalizability of the findings and suggest caution when applying these results to different settings.

-Include in the discussion the absence of a control group and the study's retrospective nature, focusing on how these factors impact the findings and their interpretation.

- The manuscript analyzes the impact of having specific carbapenemase types on outcomes. Please elaborate on how cases with multiple carbapenemases were handled in the analysis, considering their potential compound effect on treatment resistance and patient survival.

-The manuscript identifies specific factors impacting survival but could benefit from a comparative discussion with existing literature, especially where discrepancies are stated.

Please state the objectives clearly at the end of the introduction section. The objectives seem to be to describe the sources of Cr-Kpn in the hospital, assess factors associated with patient outcomes, and evaluate the effectiveness of treatments used against these pathogens. Additionally, the manuscript reports no significant difference in patient outcomes between carbapenem-based regimens and carbapenem-sparing regimens, or when comparing treatments based on ceftazidime-avibactam with other regimens.

Comments on the Quality of English Language
  • There are occasional grammatical errors that could be refined. For instance, the use of articles ("a," "an," "the") is sometimes inconsistent. Attention to subject-verb agreement and correct use of tense.
    Before submission, the manuscript would benefit significantly from professional language editing to polish grammar, syntax, and style, ensuring it meets the high standards expected in academic publishing.
    Other minor comments: please italicize all "bla" prefix before gene names. and use subscript appropriately.
  1.  
  2.  

Author Response

Thank you for your suggestions. We attached our answers.

Reviewer 2 Report

Comments and Suggestions for Authors Dear Authors, 

From my review, I make the following observations:

Title: Would be helpful to include the location where the study was conducted.

Introduction: The purpose of the search is not well defined.

Materials and methods: I would insert the materials and methods paragraph before the results. 

I will include where the study is conducted (City, Hospital, laboratory).
Clarify how the sampling was performed and by whom (trained personnel?).
Clarify the practical part of the culture process (culture media etc.).

Results: Slightly dispersive treatment and figures to be improved (some numbers do not read well with color contrast); Useful to recheck numbers and percentages.

Clarifying how many of these patients were first-time or known patients with previous infection, whether they were transferred from other wards, from home or from other care facilities.

Discussions: Enter the discussion treatment, talk about comparisons with other national and international analyses.

Talk about possible ways to solve the problem.

Include the study's limitations and strengths. Comments on the Quality of English Language

 Minor editing of English language required

Author Response

(The authors gave the same response as above.)

Reviewer 3 Report

Comments and Suggestions for Authors

The manuscript addresses an important and topical issue.  Although it contributes valuable insights into the management and understanding of carbapenem-resistant Klebsiella pneumoniae infections, I suggest revision aim to enhance clarity, detail, and the manuscript's overall impact without necessitating extensive additional research or data collection in the following points:

Title: Clearly indicates the focus on NDM and OXA 48-like carbapenem-resistant Klebsiella pneumoniae. However, it could be more precise by specifying the study's nature, e.g., a retrospective analysis.

Abstract: Provides a concise overview of the study, including background, methods, key findings, and conclusions. It could be improved by briefly mentioning the study's significance and potential implications for clinical practice or public health policy.

Introduction:

Clarity and Relevance: The introduction effectively sets the context for the research, highlighting the public health problem posed by carbapenem-resistant Klebsiella pneumoniae (Cr-Kpn). References to previous studies establish a solid foundation. However, it could benefit from a more direct statement of research gaps this study aims to address.

Methods

Study Design: The retrospective observational cohort study design is appropriate for the research objectives. However, detailing the criteria for selecting patient records and any exclusion criteria would enhance reproducibility.

Data Collection and Analysis: Describes data collection comprehensively. The statistical methods used for analysis are suitable, but the manuscript could benefit from more detail on handling missing data and any adjustments for confounding factors.

Objectives: Clearly defined, but integrating a hypothesis or specific research questions could enhance clarity regarding the study's focus.

The section between lines 453-455 needs more detail (manufacturer, product, what it was used for): "Screening of bacterial strains for membership in a particular antibiotic resistance phenotype was performed using chromogenic media, with results then confirmed by standard methods."

Results

Presentation of Data: Results are presented clearly, with appropriate use of tables and figures. However, discussing the statistical significance of findings in the context of clinical relevance would be beneficial.

Interpretation: The authors accurately interpret their findings, but a more in-depth discussion comparing their results with existing literature would provide context for the study's contributions to the field.

Discussion

Strengths and Limitations: The discussion acknowledges the study's limitations, which is good practice. Expanding on how these limitations could impact the findings and suggesting directions for future research would strengthen this section.

Implications: The authors discuss the implications of their findings for managing infections caused by Cr-Kpn. Further elaboration on potential policy or practice changes could enhance the manuscript's impact.

Conclusion

Summary of Findings: Concise and directly linked to the study objectives. Highlighting the novel contributions of the study and potential for future research could enhance the conclusion.

Specific Recommendations for Revision

It is essential to improve the quality of the figures. The Fig. A1 and A2 should be put on the same figure and bring the plots in sync (e.g. both parts of the figure should have the sensitivity percentages on top).

The quality of the tables also needs to be improved.

In line A165, in the next sentence, it would be advisable to change the subject from we to they:

"We were infected significantly with carbapenemase-producing Klebsiella pneumoniae, suggesting the important circulation of this type of bacteria in the environment."

In line 215, NDM should be written in capital letters throughout.

In line 355, replace "Gram-negative bacteria carriage" with the following:

"Coinfection and/or carriage of Gram-negative bacteria"

In the sentence between lines 372 and 374, it was not entirely clear which data referred to ceftazidime-avibactam, as it was mentioned twice:

"In our cohort, we found higher levels of sensitivity to ceftazidime-avibactam of 44.3%, colistin of 41.4%, and for NDM+OXA48-like, sensitivity to antibiotics was lower, meeting more significantly with colistin with 32%, Ceftazidime-avibactam showed a sensitivity of 8%."

Comments on the Quality of English Language

The manuscript demonstrates a competent use of scientific English appropriate for its intended academic audience. Minor improvements in language use and structure could further enhance its clarity and impact.

Author Response

(The authors gave the same response as above.)

Reviewer 4 Report

Comments and Suggestions for Authors

The present study represents the characterisation of carbapenemase resistant Klebsiella pneumoniae studied over 22 months in a single ? hospital. Despite this being a retrospective cohort study, the authors have tried to extract and present most of the desired information regarding the resistance patterns of isolates, clinical outcomes etc.  Few observations from my side which need to be addressed are as mentioned below:

1.     Abstract: Please write full forms at the first appearance e.g. NDM, OXA-48.

2.     Introduction:

Please mention the ESKAPE organisms in parenthesis with ESKAPE at first instance.

The authors need to mention the scientific rationale and aim of the present study?

3.     Material and methods:

Please describe the study setting.

The authors mention “where all patients were hospitalised and presented carbapenemase-producing Klebsiella pneumoniae isolates”; I guess this is a grammatical error and should be written as “where all hospitalised patients having documented infection with carbapenemase-producing Klebsiella pneumoniae were included”. Please check.

Please define the eligibility criteria clearly. For example, in results it is mentioned that patients who died within 72 hours of hospitalisation were excluded; this should come under methods section.

4.     Results

Table 1: What criteria were used for assessing disease severity?

Table 1: What is meant by DZ?

Table 1: A t some places, the values denote y (%) as written; what does that mean? I guess it should be n (%) as the data represents numbers.

What is portaj in table 1?

I would suggest adding the full names of antibiotics (figure 1) under abbreviations section.

The text in table A4 needs adjustment.

5.     The authors give a passing reference to One Health approach to limit the bacteria spread; would request to elaborate this a bit under discussion as well.

6.     The title refers to “ from menacing superbug to a mundane bacteria”, please justify this considering the study findings.

Comments on the Quality of English Language

Moderate editing of English language required.

Author Response

(The authors gave the same response as above.)

Round 2

Reviewer 2 Report

Comments and Suggestions for Authors

Dear Authors

Thank you for your work, which is very interesting, I think it would be useful to add a passage in the discussions or conclusions that talks about the contrast techniques and new technologies being developed.

I attach some articles that might help in this regard:

- Amodeo D, Manzi P, De Palma I, et al. Efficacy of Violet-Blue (405 nm) LED Lamps for Disinfection of High-Environmental-Contact Surfaces in Healthcare Facilities: Leading to the Inactivation of Microorganisms and Reduction of MRSA Contamination. Pathogens. 2023;12(11):1338. Published 2023 Nov 10. doi:10.3390/pathogens12111338.

- Amodeo D, Lucarelli V, De Palma I, et al. Efficacy of violet-blue light to inactive microbial growth. Sci Rep. 2022;12(1):20179. Published 2022 Nov 23. doi:10.1038/s41598-022-24563-1.

Comments on the Quality of English Language

Minor editing of English language required